# Digital Leadership Skills and Associations with Psychological Well-Being

**DOI:** 10.3390/ijerph16142628

**Published:** 2019-07-23

**Authors:** Sabrina Zeike, Katherine Bradbury, Lara Lindert, Holger Pfaff

**Affiliations:** 1Institute of Medical Sociology, Health Services Research, and Rehabilitation Science (IMVR), The University of Cologne, Medical Faculty, 50933 Cologne, Germany; 2Department of Psychology, Faculty of Social and Human Sciences, University of Southampton, Southampton SO17 1BJ, UK

**Keywords:** digital leadership, psychological well-being, WHO-5, health at work, digital transformation, digitalisation

## Abstract

Due to increasing digitalisation, today’s working world is changing rapidly and provides managers with new challenges. Digital leadership is an important factor in managing these challenges and has become a key concept in the discussion about what kinds of skills managers need for digital transformation. The main research question our study explored was if digital leadership is associated with psychological well-being in upper-level managers. Based on a qualitative pilot study and relevant literature, we developed a new scale for digital leadership in managers. We conducted an online survey with a sample of 368 upper-level managers from a large German ICT-company. Using a stepwise logistic regression analysis, potential effects of digital leadership on psychological well-being (WHO-5) were analysed. Logistic regression analyses showed that better skills in digital leadership were significantly associated with higher well-being. Results also showed that gender, age and managerial experience had no effect in our model. Our study provides a valuable insight into the association between digital leadership and well-being in managers. However, further research is necessary to validate the newly developed scale for digital leadership and to confirm a causal effect in the relationship between digital leadership and well-being.

## 1. Introduction

Companies in all industries are increasingly becoming digitalised and re-organised into new forms of digital organisations. Digital transformation is rapidly and fundamentally changing existing businesses and it is widely acknowledged that companies that miss the trend of digitalisation today will be in the future slower, less flexible and less competitive than digital pioneers [1]. Previous research has shown that digital transformation has fundamental effects on businesses, societies and individuals [1,2,3,4,5]. Technological development and job redesign continue to change work in many ways [6]. Digitalisation may, in this context, change the nature of work as well as job demands and resources [7]. Digital transformation processes are considered to be a prime challenge for leadership and top management of transforming organisations [1,8,9]. Westerman et al. (2014) showed that companies “that struggle with becoming truly digital fail to develop digital capabilities to work differently and the leadership capabilities required to set a vision and execute on it” [1] (p. 3). In contrast, the authors claim that digitally successful companies have built strong leadership capabilities to envision and drive transformation. In this context, leadership capabilities are the ways in which managers are driving change. Larjovuori et al. (2016) defined digital leadership as “the leaders’ ability to create a clear and meaningful vision for the digitalization process and the capability to execute strategies to actualize it” [7] (p. 1144). According to a competence model for digital leaders, two dimensions that make up a successful digital leader can be distinguished: (1) attitudes, competences and behaviours that managers need in the digital age (e.g., digital literacy/competences) and (2) competencies that help drive digital transformation (e.g., strong leadership skills) [10,11]. To successfully master the challenges of digital transformation, it is important to strengthen crucial job resources in managers (i.e., digital leadership skills/capabilities). Excessive job demands have been, when not accompanied by adequate job resources, shown to be associated with reduced well-being and higher risks of burn-out [12,13]. Previous studies have shown that managers often experience high degrees of distress [14,15,16].

Although digitalisation and its consequences are often a matter of debate, the actual implications on work-related health have not yet been researched very well. This applies in particular to the potential effects of digital transformation on managers’ health. The aim of this cross-sectional study was to test associations between digital leadership skills of upper-level managers and perceived psychological well-being. We assumed that having good digital leadership skills could serve as a job resource and may positively influence managers’ well-being. Our underlying research question was: Is digital leadership associated with psychological well-being in upper-level managers and, if so, how strong is the influence? 

During the literature review process, we found that there are various terms for digital literacy (e.g., computer literacy, ICT-literacy, digital competence, digital readiness) and digital leadership (e.g., digital leadership skills/capabilities/abilities), which makes it challenging to present a bigger picture of previous research. We also found that digital leadership is an emerging field with few, if any, theory-based and/or empirically validated concepts and instruments. We found the competence model for digital leaders (as mentioned above), developed by Capgemini Consulting (2015) [10], and a questionnaire assessing digital leadership capabilities from the CEO’s point of view, developed by Westerman et al. (2014) [1], particularly interesting for our study. However, we could not find a validated scale assessing self-perceived digital leadership skills in managers; thus, we developed a new scale for the purpose of our study. The development was based on a qualitative pilot study and previous research and concepts. The previous research is primarily from management literature [1,7,10,11,17]. However, especially in the field of public health, concepts on digital leadership are still seldom used. 

## 2. Theoretical Background

The Job Demands-Resources (JD-R) model from Bakker and Demerouti provides a broader theoretical framework for our study [12]. According to the model, excessive job demands are, when not accompanied with adequate job resources, associated with reduced health outcomes.

There are consistent findings that job demands and work-related stress are related to specific health outcomes such as sleep disturbances [18,19] and cardiovascular risk [20,21,22]. Managers especially are often exposed to high levels of job demands such as strong pressure to perform at a high level and having to meet deadlines [23]. Furthermore, they are confronted with the job of simultaneous supervision of various tasks, frequent interruptions and role overload [23,24]. Studies have also shown that managers do often experience high degrees of distress [14,15,16,25].

Despite the association between job demands and negative health outcomes, such as reduced psychological well-being, only a few studies have explored the working conditions of managers and especially specific job resources in the context of digitalization.

In the present study, we focused on job resources and specifically on digital leadership skills as an internal job resource of upper-level managers. In this context, internal job resources are understood as resources that affect the person and their psychological characteristics and competences as well as their physical characteristics. Well researched as internal job resources are personality traits such as control beliefs, perceived self-efficacy, optimism and intelligence, but also skills such as coping skills or social skills [26]. Furthermore, previous studies confirm that health literacy serves as a job resource and positively influences well-being in managers [16]. Based on these findings, we assumed that digital leadership also serves as a job resource and positively influences psychological well-being in managers. We hypothesized that managers need specific digital leadership skills to positively manage the challenging demands of driving digital transformation processes and that these—according to the JD-R-model—are positively associated with psychological well-being. As far as we know, this study is the first study to test this association.

Furthermore, this study is based on the concept of psychological well-being by the World Health Organization (WHO). More recently, the WHO has defined psychological/mental well-being as “a state of well-being in which the individual realises his or her own abilities, can cope with the normal stresses of life, can work productively and fruitfully, and is able to make a contribution to his or her community” [27]. Psychological well-being is a multidimensional concept. It includes aspects of self-esteem and satisfaction with life and is about lives going well. The concept of psychological well-being covers the combination of “feeling good and functioning effectively” [28] (p. 137). The concept of effective functioning includes the development of one’s own potential, the existence of control over one’s own life, meaningfulness (e.g., working towards valuable goals) and positive relationships. External circumstances (e.g., work and working conditions [29,30,31,32]) affect our well-being, but our actions and attitudes also have a great influence [28]. Studies have shown that poor psychological well-being is a signal of distress and an indication of possible depression [33,34]. 

## 3. Materials and Methods 

The data were based on a cross-sectional survey study, which lasted from June to July 2017, and took place at a large German ICT-company. At the time of the study, extensive transformation processes were taking place at the company (e.g., reorganisations, implementation of new technologies). Five expert interviews were conducted as a pilot study to obtain expert views on the topics of interest: i.e., current digital transformation in the company and implications for managers’ workload and well-being. On the basis of the results, the questionnaire was developed and afterwards pretested within our target group, using cognitive interviews. Participation in the survey was voluntary. The data were anonymously collected and analysed. Participants gave their consent for the survey, and permission to analyse all information from the questionnaire and publish it in an anonymised form for research purposes. The study design and realisation were presented to the Ethics Committee of the University of Cologne. No objections to any aspects of the study were raised.

### 3.1. Study Design and Participants

Data were collected using a web-based survey tool (LimeSurvey) (LimeSurvey GmbH, Hamburg, Germany). The study was supported by the company’s chief human resources officer (CHRO). All upper-level managers (*N* = 1760) were invited to participate in the survey. Upper-level managers, in this context, are executives who are responsible for managers in lower management. Managers at this level have high responsibilities and must define what kinds of goals should be achieved. Because digital transformation is considered to be a prime challenge for upper management, we considered this sample to be extraordinarily interesting.

The CHRO encouraged the managers to participate in the survey. He informed the participants that their privacy would be protected, explained the procedure, and highlighted the possible benefits of the survey. The total design method (TDM) by Dillman was used as a general framework for designing the email survey [35]. Four e-mails were sent out by the company’s CHRO. The first e-mail notified participants before the start of the survey and the second e-mail was sent out at the beginning of the survey. In addition, two reminders were sent out at intervals of one week to increase the response rate. The questionnaire was available in English and German, in order to reach all executives.

Figure 1 presents the selection of the sample. A total of 1760 upper-level managers were informed about the survey. Two managers sent e-mails stating that they no longer performed any leadership tasks. Eight managers sent e-mails declining participation for various reasons. A total of 1175 managers did not participate in the survey and did not respond to any of the e-mails. Ultimately, 575 managers filled in the questionnaire. A total of 368 upper-level managers completed more than 70% of the questionnaire and were included in the analysis sample (response rate: 20.9%). Sample characteristics are shown in Table 1. The study population has also been described elsewhere [36]. In Table 2 correlations between all study variables are shown.

### 3.2. Measures

#### 3.2.1. Dependent Variable

In this study, we used the German and English versions of the WHO-5 well-being index as our primary outcome measure [37,38]. The five-item questionnaire of WHO-5 is a self-administered questionnaire measuring current psychological well-being and is among the most widely accepted questionnaires assessing subjective psychological well-being. The WHO-5 questionnaire has a total of five simple and non-intrusive questions about subjective psychological well-being. The tool has been validated as a sensitive and specific screening tool for depression [34]. The scale was first published in 1998 and has been translated into 30 languages and used all over the world [16,34,36,39,40,41]. The WHO-5 covers five positively worded items, related to positive mood, vitality and general interests in a time frame encompassing the previous two weeks. Each of the items is rated on a six-point Likert scale from 0 (not present) to 5 (constantly present). Scores are summated, with the raw scores ranging from 0 to 25. Internal consistency was Cronbach’s α = 0.87 for the present study. Cut-off scores indicating poor or high psychological well-being are well-established for the WHO-5 [34]. A raw score below 13 indicates poor well-being and is an indication for testing for depression under ICD-10 (International classification of diseases) [33]. 

#### 3.2.2. Independent Variable

Because we could not find a suitable and validated scale for digital leadership skills in upper-level managers, we developed a new scale for the purpose of our study. The development of the instrument was guided by a literature review and the findings of a qualitative pilot study. The newly developed scale was based on a competence model, developed by Capgemini Consulting (2015) [10]. According to the model, two dimensions that make a successful digital leader can be distinguished: (1) attitudes, competences and behaviour needed in digital working environments (e.g., adequate skills to use technology, good digital literacy (items 1–3 of our scale), and (2) a clear vision of digital transformation processes and capabilities to use and actualise digital strategies (items 4–6 of our scale) [10]. The development of our scale for digital leadership was based on these two dimensions. We further used the framework of ‘digital literacy’ by Health Education England (HEE) for dimension one and concepts of strategic leadership/digital leadership by Westerman and Larjovuori et al. for dimension two [1,7,42]. The framework of HEE defines digital literacy as the “capabilities that fit someone for living, learning, working, participating and thriving in a digital society” [42] (p. 2). Westerman et al. (2014) described what capabilities leaders need in order to drive digital transformation and Larjovuori et al. (2016) provided a useful definition of digital leadership (see Chapter 1). The new scale consists of six items, assessing attitudes, competences and behaviour in the context of using digital tools (items 1–3) and assessing digital leadership skills (items 4–6): (1) “I think using digital tools is fun”, (2) “I would say I am a digital expert”, (3) “When it comes to digital knowledge, I am always up to date”, (4) “I am driving the digital transformation forward proactively in our unit”, (5) “I can make others enthusiastic about the digital transformation”, (6) “I have a clear idea of the structures and processes that are needed for the digital transformation”. Each of the items had to be answered on a five-point Likert scale ranging from 1 “disagree completely” to 4 “agree completely”. After developing the scale, it was discussed and refined by a team of six experts from different occupations (ICT-specialists, occupational health specialists and specialists in questionnaire development). The questionnaire was pretested in cognitive interviews to ensure the survey met the purpose of our study, to avoid problems with comprehension and to test for face validity [43]. The questionnaire was then translated into English by a native speaker. For the present study, internal consistency was Cronbach’s α = 0.87. A confirmatory factor analysis showed a clear two-factor structure of our scale.

#### 3.2.3. Confounding Variables

Previous research has shown that well-being is a complex and multidimensional concept with differences in gender and socio-economic status [44]. We therefore considered several variables that could plausibly have confounding effects on our analysis. Potential confounders included: gender, age and managerial experience. In our study, gender was dichotomised as male or female. Age was measured in five categories (<30 years; 31–40 years; 41–50 years; 51–55 years; >55 years). Managerial experience was assessed in full years. Previous research suggests that there are generational differences between digital natives and digital immigrants, especially in the use of new technology [45]. We therefore assumed that age could have a confounding effect in our model. Gender and years of leadership experience were also considered to have a confounding effect in our model and were added explorative.

### 3.3. Statistical Analysis

According to the cut-off score of <13 for the WHO-5 well-being index, responses for psychological well-being were scored and dichotomised into groups of high and low well-being. Chi-square and t-tests were conducted to test the hypothesis of equal means between the groups of low and high psychological well-being. A logistic regression analysis was performed to test for associations between well-being and digital leadership skills. Years of managerial experience and digital leadership skills were used as continuous variables. Age was used as categorical variable. 

In Model 1 of our analysis we tested the unadjusted effects of all variables (crude analysis). In Model 2 we tested the effect of our independent variable on the dependent variable, adjusted for confounding variables (see 2.2.3). Odds ratios (OR), the p-value, their corresponding 95% confidence intervals (CI) and Nagelkerke’s pseudo-R2 were calculated. Data from managers who did not finish at least 70% of the questionnaire were excluded from the analysis (*n* = 207; see Figure 1). No missing values were imputed. All statistical analyses were performed using SPSS 25 (SPSS Inc., Chicago, IL, USA) and R version 3.5.2 (R Foundation for Statistical Computing, Vienna, Austria). A *p*-value of less than 0.05 was considered statistically significant.

## 4. Results

### 4.1. Sociodemographic Characteristics of the Participants

A total of 368 managers participated in the study. Of these, 14 managers (3.8%) filled in the English version and 354 managers (96.2%) filled in the German version of the questionnaire. Table 1 presents the descriptive characteristics of the study participants. Table 2 shows the Pearson correlation r for all study variables. Results show that well-being is significantly correlated with digital leadership (r = 0.28, *p* < 0.01). Furthermore, results show that digital leadership is significantly correlated with gender (r = 0.18, *p* < 0.01) and age (r = −0.13, *p* < 0.05). Male and younger managers rated their skills significantly better. In Table 3, the averages for gender and age of our analysis sample and the population of upper-level managers in the company are shown. The juxtaposition shows that the two groups are comparable. The maximum deviation is 5.5% for the age group 51–55 (see Table 3). Of all participants, 76.9% were male and 23.1% were female. The average managerial experience was 11.5 years, with a standard deviation of 6.73 and a range from 0 to 50 years. Most upper-level managers of our sample were responsible for 10–99 managers in lower management (57.2%; *n* = 191). Only 4.2% of the surveyed managers were responsible for more than 1000 managers in lower management (see Table 3).

The average score for psychological well-being was 15.73, with a standard deviation of 4.6 and a range from 0 to 25. The findings show that 21.5% of the surveyed managers were classified as having poor well-being (*n* = 72) and 78.5% had high well-being (*n* = 263). The average of perceived digital leadership among all participants was medium to high (M = 17.61, SD = 3.78). Bivariate comparisons between the groups of low and high well-being using a t-test revealed a *p*-value = 0.000 for digital leadership; thus, we can assume that differences in digital leadership skills between groups of high and low well-being are significant. For managerial experience, we found no evidence of equal means between the groups (*p*-value = 0.695). In a chi-square test, we found no significant results for gender (*p*-value = 0.256) and age (*p*-value = 0.404). We can assume that whether someone is in a group of low or high well-being is not dependent on both variables (see Table 4). 

### 4.2. Associations Between Digital Leadership and Psychological Well-Being

In Table 5, the results of the association between well-being and digital leadership, based on the logistic regression analysis, are summarised. Model 1 of the multivariate analysis shows the unadjusted model (crude analysis). Model 2 shows the adjusted model for the covariates gender, age and managerial experience (see Table 5). Nagelkerke’s R square is 0.127 for the adjusted model. According to Cohen (1992), this corresponds to a strong effect [46]. The results show that better skills in digital leadership are significantly associated with higher psychological well-being (*p* = 0.000, OR = 3.11, 95% CI = 1.93–4.99). The results also show that the confounding variables gender, age and managerial experience had no effect in our model. From our analysis, that means that whether someone has a low or high psychological well-being cannot be predicted by our confounding variables.

## 5. Discussion

For the purpose of this study, we developed a scale for digital leadership, based on previous research and leadership concepts. Our findings provide evidence of an association between perceived digital leadership skills and psychological well-being in upper-level managers. In line with our hypothesis, we found that managers with lower digital leadership skills are more likely to have low psychological well-being. To our knowledge, this study is the first to analyse this association.

Previous studies have shown that a score for well-being below 13 is a first indication for depression and denotes “mental health at risk” [34,47]. In our study, 21.5% of the surveyed managers were classified with well-being below that threshold. These findings are comparable to results from other studies investigating psychological well-being using the WHO-5. The Fifth European Working Conditions Survey has shown that 19.5% of managers across the EU report a score below 13 for psychological well-being [48]. In a study by Fiedler et al., 25% of surveyed managers were classified as having low psychological well-being [16]. These quite high levels support the growing need for attention to be paid to mental health in the workplace, especially in times of digitalisation and new challenges for leadership.

The European Working Conditions Survey also revealed gender differences: in 15% of men and 24% of women, well-being was found to be at risk. In our study, no gender differences were found, which could be due to the fact that the majority of our participants were men (76.9%). This can be explained by the fact that our study took place in the ICT-industry, where the majority are still men. However, our sample is comparable to other studies in this sector [16]. To date, research on digital leadership has been mainly in the management field: e.g., what characteristics are important for digital leaders and how these relate to the success of transformation/change processes or performance [1,49,50]. Studies from this research field have shown that companies with high leadership capabilities are more successful than companies which do not invest in digital leadership [1,10,11]. In addition to these findings, our study was able to show that digital leadership is associated with managers’ well-being. The findings of our study also show that gender, age and managerial experience had no effect on this association. Because no effects were found for our confounding variables, we can deduce that the results of our study are relevant for all genders, ages, and all managers, regardless of leadership experience. A representative study by Boehm et al. (2016) supports our findings for age and shows that age-related differences in technology optimism, technological skills and fear of job loss due to technology are rather small [51].

### 5.1. Strengths and Limitations

This study has some strengths and limitations, which should be mentioned. The key limitation of our study is the cross-sectional design. Causal conclusions and developments over time are not possible on the basis of cross-sectional data [52]. Further studies with longitudinal designs are necessary to confirm our findings. Longitudinal studies can provide stronger evidence of the directionality of the hypothesized relationship between digital leadership and well-being. Furthermore, all observations were based on self-reports, which can cause effects to be over- or underestimated [53]. The scale for digital leadership was newly developed in our study and has not yet been scientifically validated. However, the scale showed good reliability (internal consistency: α = 0.87). Because no validated measures could be found for digital leadership skills in managers, testing concurrent validity was not possible. 

One further limitation of this study is that the models do not include a measure of work-related stress, which could also be related to well-being. The focus of this study was primarily on digital leadership. However, in further studies work-related stress should also be considered as a potential confounding variable. In our study, we collected data only in one large German company in the ICT-sector. The study sample of our study might therefore not be representative for managers in general. We must therefore assume that this limits the generalizability of our results. However, the study sample comes from a large internationally cooperating company and was carried out at several locations throughout Germany. Company cultures and structures vary a lot, but it can be assumed that there are general processes working here, especially when it comes to digital transformation processes, that are not specific to only the one company in the sample but can also be expected to be observed in other companies with similar structures. Further research is necessary to test this assumption.

In addition, the response rate was relatively low (20.9%). However, the comparison of our analysis sample with the population sample in the company showed little difference between the groups. Reasons for the low response rate may have been that our survey took place during summer, which is when many people take vacation, and the run time of our survey was relatively short.

### 5.2. Implications for Practice and Further Research

Our study provides the first evidence of an association between digital leadership skills and well-being among managers. However, further longitudinal studies are needed to confirm this association over time. As a practical implication, our research shows that digital leadership skills might be important to well-being. Based on our findings, further research should investigate whether improving digital leadership skills improves well-being in managers. Further research could also demonstrate whether improving digital leadership skills has an impact on stress. This might be the case, because work stress is associated with reduced well-being [54]. In addition, further research will be needed to validate the scale and to establish if there is a causal mechanism in the relationship between digital leadership and well-being. Research is also needed to confirm the direction of the relationship between digital leadership and well-being, because it could also be assumed that managers with lower psychological well-being are more pessimistic about their digital leadership skills and assess themselves more poorly than managers with good psychological well-being (reverse causation).

## 6. Conclusions

The role of leadership in digital transformation processes is an emerging field of research. In our study, we found evidence that upper-level managers with lower digital leadership skills are more likely to have low psychological well-being. To our knowledge, the present study was the first to analyse this association. However, further research is needed to validate the newly developed scale for digital leadership and to confirm whether there is a causal effect of digital leadership on psychological well-being.

## Figures and Tables

**Figure 1 ijerph-16-02628-f001:**
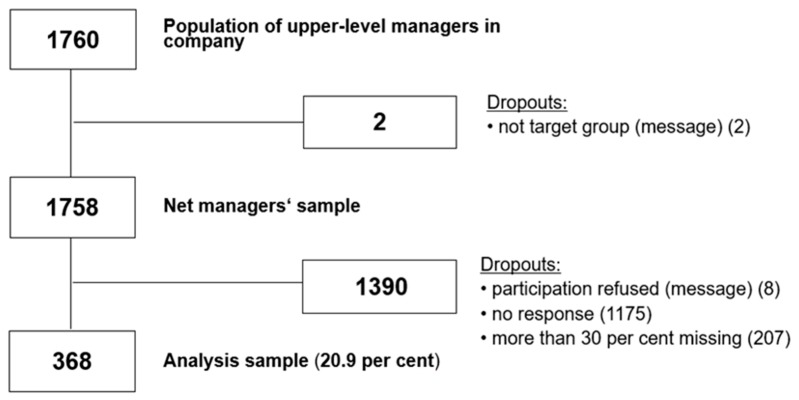
Flowchart of the selection of the managers’ sample.

**Table 1 ijerph-16-02628-t001:** Descriptive characteristics for all model variables.

Variable	*N*	M	SD	Median	Minimum	Maximum
Digital Leadership	335	17.61	3.78	18	7	24
WHO-5	335	15.73	4.60	16	0	25
Managerial experience (in full years)	334	11.50	6.73	10	0	50
**Variable**		**Response Trait**		**Frequency (*n*)**		**Percentage**
WHO-5 (dichotomised)	335	Low (<13)		72		21.5
		High (≥13)		263		78.5
Gender	334	Male		257		76.9
		Female		77		23.1
Age	334	<30		0		0.0
		31–40		13		3.9
		41–50		157		47.0
		51–55		126		37.7
		>55		38		11.4

**Table 2 ijerph-16-02628-t002:** Correlations between the variables.

Variable	*N*	1	2	3	4	5
(1) Well-being (WHO-5)	335	(0.87)	0.28 **	−0.06	0.03	−0.01
(2) Digital Leadership	312	0.28 **	(0.87)	0.18 **	−0.13 *	−0.06
(3) Gender	334	−0.06	0.18 **	-	0.06	0.08
(4) Age	334	0.03	−0.13 *	0.06	-	0.45 **
(5) Managerial experience	334	−0.01	−0.06	0.08	0.45 **	-

Notes: Pearson correlation r and α values (in the diagonal) are shown; * *p* < 0.05; ** *p* < 0.01; Gender had the values of 1 (women) and 2 (men).

**Table 3 ijerph-16-02628-t003:** Comparison analysis sample to population of upper-level managers in ICT-company.

Variable	Analysis Sample; *n* = 334 % (*n*)	Population; *n* = 1760 % (*n*)
Gender		
men	76.9 (257)	81.6 (1437)
woman	23.1 (77)	18.4 (323)
Age		
<30	0 (0)	0 (0)
31–40	3.9 (13)	8.0 (140)
41–50	47.0 (157)	44.9 (790)
51–55	37.7 (126)	32.2 (567)
>55	11.4 (38)	14.9 (263)

**Table 4 ijerph-16-02628-t004:** Descriptive statistics of the independent variables for managers with high and low psychological well-being.

Variable	Managers with High Well-Being				Managers with Low Well-Being				t-Test *p*-Value
	N	M	SD	Median	N	M	SD	Median	
Digital Leadership	244	18.08	3.59	18.00	68	15.65	4.01	15.00	0.000
Managerial experience	262	11.42	6.34	10.00	72	11.82	8.05	10.00	0.695
**Variable**		**Managers with High Well-Being Percentage**			**Managers with Low Well-Being Percentage**				**Chi-SquareTest** ***p*-Value**
Gender									0.256
Men		75.6			81.9				
Women		24.4			18.1				
Age									0.404
<30		0			0				
31–40		3.8			4.2				
41–50		48.5			41.7				
51–55		35.5			45.8				
>55		12.2			8.3				

**Table 5 ijerph-16-02628-t005:** Results of the logistic regression analysis.

Variable	Model 1 Unadjusted Model (Crude Analysis)			Model 2 Adjusted Model		
	OR	95% CI	*p*	OR	95% CI	*p*
Digital Leadership	2.84	1.80–4.50	0.00	3.11	1.93–4.99	0.00
Gender	0.68	0.35–1.32	0.26	0.60	0.30–1.23	0.17
Age						
31–40 (1)	0.63	0.13–2.97	0.55	0.54	0.08–3.75	0.54
41–50 (2)	0.79	0.30–2.07	0.64	0.59	0.20–1.79	0.35
51–55 (3)	0.53	0.20–1.38	0.19	0.45	0.16–1.29	0.14
>55 (reference)						
Managerial experience	0.99	0.95–1.03	0.65	1.00	0.95–1.04	0.83
Cox & Snell R Square						0.082
Nagelkerke’s pseudo-R Square						0.127

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
