# Peer review of "Digital Leadership Skills and Associations with Psychological Well-Being"

_ijerph, 2019, doi:10.3390/ijerph16142628_

Round 1

Reviewer 1 Report

in the lines 78-79-80, the author explain the theoretical model of JD-R and digital leadership but they don't consider any literature support to motivate the link between the model and the leadership. why digital leadership is considered a job resource? The authors could better specify this issue.

Author Response

Response to Reviewer 1 Comments

Point 1: In the lines 78-79-80, the author explain the theoretical model of JD-R and digital leadership but they don't consider any literature support to motivate the link between the model and the leadership. why digital leadership is considered a job resource? The authors could better specify this issue.

Response 1:

We are grateful to reviewer 1 for taking the time to make constructive criticism on how this manuscript could be improved. We agree with the comment and have now focused on this aspect in more detail. We added more information about this issue (see p. 2 & 3, lines 80-100).

Reviewer 2 Report

·         Summary of the manuscript

The purpose of this manuscript is to investigate whether digital leadership is associated with psychological well-being for upper-level managers. The authors developed a new scale for managers’ digital leadership and conducted an online survey for 368 upper-level managers from German ICT companies. Main finding is that digital leadership shows a significant association with psychological well-being different from gender, age, managerial experience, and managerial responsibility. Although digitalization is an important factor in today’s business, the current research was designed in an improper way and the contribution is too marginal to be published in current form.

·         General comments

- First, the reviewer is disappointed that the data were collected at ONE large German ICT company. It must generate a huge doubt about the generalizability of the finding/insight of the research. Although 368 samples look enough to conduct statistical tests, it may cause a controversy about whether the result in a company can be applied to other companies. The reviewer believes that there exists a better research design and method for this research and suggests to conduct a qualitative case study rather statistical tests.

- Second, the reviewer believes that Introduction section can be improved by summarizing the main findings in that section. Now the section contains both the research question and what the authors have done. However, the section does NOT tell the readers about what is the main finding of the research. Thus, the reviewer suggests to rewrite Introduction section with the main findings of the research.

- Third, the reviewer believes that Theoretical Background section must be improved into a more systematic way. The authors adopted the well-known Job Demand-Resource (JD-R) model and claimed that ‘digital leadership skill is a job resource.’ If the authors want to insist that the JD-R model is adequate for the research and that digital leadership skill is a job resource, then theoretical background must be reinforced.

- Fourth, the reviewer wonders that assessing ‘Managerial Responsibility’ based on the number of employees can be acceptable. Is there any reference measuring the managerial responsibility in such a way? Or this scale is also developed by the authors?

Author Response

Response to Reviewer 2 Comments

Point 1: First, the reviewer is disappointed that the data were collected at ONE large German ICT company. It must generate a huge doubt about the generalizability of the finding/insight of the research. Although 368 samples look enough to conduct statistical tests, it may cause a controversy about whether the result in a company can be applied to other companies. The reviewer believes that there exists a better research design and method for this research and suggests to conduct a qualitative case study rather statistical tests.

Response 1:

Thank you very much for the comment. We agree that the generalizability of our findings is limited because of the study design (cross-sectional, single source). We discussed this issue in our limitations section (see. p.10, lines 322-338).

However, our study was conducted at the highest management level of one of the 30 major German DAX companies. Access to this target group is not always easy for conducting research. We are therefore convinced that our dataset is of great value for science and believe – despite the methodological limitations – that our study represents an added value for research. Digital leadership is still largely unexplored, and our study provides initial evidence of how new demands on leaders are impacting mental well-being. In the discussion chapter, we also discussed that the topic and the newly developed scale should be researched in further studies (see. p.10&11, lines 344-358).

We further agree with the reviewer that a qualitative case study would also have been appropriate.

Point 2: Second, the reviewer believes that Introduction section can be improved by summarizing the main findings in that section. Now the section contains both the research question and what the authors have done. However, the section does NOT tell the readers about what is the main finding of the research. Thus, the reviewer suggests to rewrite Introduction section with the main findings of the research.

Response 2:

Thank you very much for the advice. We summarized the main findings of our study in the abstract and discussion part. From our point of view, a further summary of the main findings in the introduction part is not required. We have decided not to implement this suggestion. We hope that the reviewer understands this.

Point 3: Third, the reviewer believes that Theoretical Background section must be improved into a more systematic way. The authors adopted the well-known Job Demand-Resource (JD-R) model and claimed that ‘digital leadership skill is a job resource.’ If the authors want to insist that the JD-R model is adequate for the research and that digital leadership skill is a job resource, then theoretical background must be reinforced.

Response 3: We fully agree with this comment. The JD-R model was only mentioned in the first version of our manuscript but not further explained. We have now expanded on this and explained why we assume in our study that digital leadership is a job resource. (see. p.2&3, lines 80-100)

Point 4: Fourth, the reviewer wonders that assessing ‘Managerial Responsibility’ based on the number of employees can be acceptable. Is there any reference measuring the managerial responsibility in such a way? Or this scale is also developed by the authors?

Response 4: We asked managers about the number of employees the respondent was responsible for, as we felt that the extent of the area of responsibility could have an impact on well-being. The variable was therefore included in the model as a confounding variable. We used the number of employees as a proxy measure for managerial responsibility. We based it on the contribution provided by the company.

Reviewer 3 Report

This cross sectional study investigated the relationship between digital leadership and psychological well-being in a sample of 368 upper-level managers. Study design is overall accurate and results are well presented and discussed. Results confirm the hypothesis that digital leadership affects psychological well-being, also considering possible confounders. 

Some suggestions:

Theoretical Background: It would be interesting that you reflect on the relationship between well-being and work related stress (see also Lecca LI, Campagna M, Portoghese I, et al. Work Related Stress, Well-Being and Cardiovascular Risk among Flight Logistic Workers: An Observational Study. Int J Environ Res Public Health. 2018;15(9):1952. Published 2018 Sep 7. doi:10.3390/ijerph15091952)

Table 1. It is not clear what is refered to. is it the correlation between variables? did you use a Pearson's or a Sperman's correlation? Please specify in the text and in the caption.

Moreover, you don't talk about correlation in the statistical analysis section. Please add this issue.

Table 2. please better specify in the text and correct the caption (I can't see the correlations in table 2.)

 Discussion: I suggest to consider the lack of work related stress assessment as a potential limitation of the study. Please discuss this aspect.

Author Response

Response to Reviewer 3 Comments

Point 1: Theoretical Background: It would be interesting that you reflect on the relationship between well-being and work related stress (see also Lecca LI, Campagna M, Portoghese I, et al. Work Related Stress, Well-Being and Cardiovascular Risk among Flight Logistic Workers: An Observational Study. Int J Environ Res Public Health. 2018;15(9):1952. Published 2018 Sep 7. doi:10.3390/ijerph15091952)

Response 1:

We are grateful to reviewer 3 for taking the time to make constructive criticism on how the manuscript could be improved. We agree with the comment and included additional information on work-stress in managers and associations with well-being. Furthermore, we extended on well-known job resources in managers and why we considered digital leadership to be a job resource (see p. 2 & 3, lines 80-100).

Point 2: Table 1. It is not clear what is refered to. is it the correlation between variables? did you use a Pearson's or a Sperman's correlation? Please specify in the text and in the caption. Moreover, you don't talk about correlation in the statistical analysis section. Please add this issue.

Response 2:

Thank you very much for your comment. Unfortunately, the tables had shifted in the document. We have now corrected the table’s captions and referred more to the tables in the text. (see Table 1 + p. 4, lines 149-150).

We now wrote in the results section (see p.6, lines 245-249):

“Table 2 shows the Pearson correlation r for all study variables. Results show that well-being is significantly correlated with digital leadership (r=0.28, p<0.01). Furthermore results show that digital leadership is significantly correlated with gender (r=0.18, p<0.01) and age (r=-0.13, p<0.05). Male and younger managers rated their skills significantly better.”

Point 3: Table 2. please better specify in the text and correct the caption (I can't see the correlations in table 2.)

Response 3: Here, too, we would like to thank you for the comment. We have corrected this (see Table 2 and p. 4, lines 149-150).

Point 4: Discussion: I suggest to consider the lack of work related stress assessment as a potential limitation of the study. Please discuss this aspect.

Response 4: Thank you very much for the advice. We agree and added information about this issue. We now added in the discussion chapter (see p. 10 lines 332-335):

“One further limitation of this study is that the models do not include a measure of work related stress, which could also be related to well-being. The focus of this study was primarily on digital leadership. However, in further studies work-related stress should also be considered as a potential confounding variable.”

Round 2

Reviewer 2 Report

The reviewer appreciates the kind response and explanation of the authors. However, the reviewer believes that the authors’ responses do NOT meet the standard of theoretical reasoning.

First, the authors agreed that there may be a better research design and methodology using the valuable dataset as the reviewer commented. However, the authors did NOT reflect the comments and limitation for generalizability remains unsolved.

Second, theoretical background for ‘digital leadership’ and ‘job responsibility’ is NOT persuasive. The only responses of the authors are ‘it is our idea/decision’. However, an idea is not a sufficient evidence for theoretical reasoning. The reviewer suggests to reinforce theoretical reasoning for constructs and scales used uniquely in this research.

Author Response

Response to Reviewer 2 Comments (round 2)

Point 1: First, the authors agreed that there may be a better research design and methodology using the valuable dataset as the reviewer commented. However, the authors did NOT reflect the comments and limitation for generalizability remains unsolved.

Response 1: Again, we would like to thank the reviewer for taking the time to make constructive criticism on how this manuscript could be improved.

We continue to believe that our study is of great value and we disagree that the study design is inappropriate. In our last reply, we just agreed that a case study also would have been possible. However, methodological limitations of the cross-sectional design and limitations related to the generalizability of our findings have been discussed in our manuscript (see p.9, lines 307-310 & 320-323).

Furthermore, we would like to reiterate that this is a valuable dataset from a large internationally cooperating company. The study was conducted throughout Germany (more than 100 locations) and a heterogeneous group of managers was surveyed (national and international, survey in German and English). It is true that company cultures and structures vary a lot, but it can be assumed that there are general processes working here that are not specific to only the one company in the sample but can also be expected to be observed in other companies with similar structures. Of course, as discussed in our manuscript, further research is necessary, to verify this assumption. We extended the issue of generalizability in our paper (see p.9 & 10, lines 323-328). If there are further concrete suggestions by the reviewer, we will be happy to extend this topic.

Point 2: Second, theoretical background for ‘digital leadership’ and ‘job responsibility’ is NOT persuasive. The only responses of the authors are ‘it is our idea/decision’. However, an idea is not a sufficient evidence for theoretical reasoning. The reviewer suggests to reinforce theoretical reasoning for constructs and scales used uniquely in this research

Response 2:

First, in our manuscript we have explained the theoretical grounding of the newly developed scale ‘digital leadership’ in detail (see introduction & chapter 3.2.2., p. 2, lines 68-75 & p. 5, lines 172-200). The scale is mainly based on a competence model for digital leaders with two dimensions (1: attitudes, competences and behaviour and 2: clear vision of digital transformation processes and capabilities to use and actualise digital strategies). We explained that our scale is based on these two dimensions and a confirmatory factor analysis confirmed the two-factor structure of the scale ‘digital leadership’. We further explained which previous research and concepts we found particularly interesting for our study (Westerman et al. 2014, Larjovuori et al. 2016, framework of ‘digital literacy’ by Health Education England).

Second, we agree that there was no theoretical reasoning for the confounding variable ‘managerial responsibility’. For that reason, we have now removed this variable from our study. We found this uncritical, as managerial responsibility was not a main topic of our paper, only a single-item (How many employees are you responsible for overall?) and not significant in our model.

We hope the reviewer finds our revised paper improved and that we have dealt satisfactorily with all comments and suggestions.